# Review of Gold Nanoparticles in Surface Plasmon-Coupled Emission Technology: Effect of Shape, Hollow Nanostructures, Nano-Assembly, Metal–Dielectric and Heterometallic Nanohybrids

**DOI:** 10.3390/nano14010111

**Published:** 2024-01-02

**Authors:** Kalathur Mohan Ganesh, Seemesh Bhaskar, Vijay Sai Krishna Cheerala, Prajwal Battampara, Roopa Reddy, Sundaresan Chittor Neelakantan, Narendra Reddy, Sai Sathish Ramamurthy

**Affiliations:** 1STAR Laboratory, Department of Chemistry, Sri Sathya Sai Institute of Higher Learning, Prasanthi Nilayam Campus, Sri Sathya Sai District, Puttaparthi 515134, India; rsaisathish@sssihl.edu.in; 2Department of Electrical and Computer Engineering, University of Illinois at Urbana-Champaign, Urbana, IL 61801, USA; 3Nick Holonyak Jr. Micro and Nanotechnology Laboratory (HMNTL), University of Illinois at Urbana-Champaign, Urbana, IL 61801, USA; 4Carl R. Woese Institute for Genomic Biology, University of Illinois at Urbana-Champaign, Urbana, IL 61801, USA; 5Department of Chemistry, Sri Sathya Sai Institute of Higher Learning, Brindavan Campus, Kadugodi, Bengaluru 560067, India; vijaysaikrishna.ch@gmail.com (V.S.K.C.); cnsundaresan@sssihl.edu.in (S.C.N.); 6Center for Incubation Innovation Research and Consultancy, Jyothy Institute of Technology, Thataguni Post, Bengaluru 560109, India; shettyprajwal755@gmail.com (P.B.); roopa.reddy@ciirc.jyothyit.ac.in (R.R.); narendra.r@ciirc.jyothyit.ac.in (N.R.)

**Keywords:** surface plasmon-coupled emission (SPCE), gold nanoparticle (AuNP), photonics, biosensing, nano-assembly, metal–dielectric, nanohybrid

## Abstract

Point-of-care (POC) diagnostic platforms are globally employed in modern smart technologies to detect events or changes in the analyte concentration and provide qualitative and quantitative information in biosensing. Surface plasmon-coupled emission (SPCE) technology has emerged as an effective POC diagnostic tool for developing robust biosensing frameworks. The simplicity, robustness and relevance of the technology has attracted researchers in physical, chemical and biological milieu on account of its unique attributes such as high specificity, sensitivity, low background noise, highly polarized, sharply directional, excellent spectral resolution capabilities. In the past decade, numerous nano-fabrication methods have been developed for augmenting the performance of the conventional SPCE technology. Among them the utility of plasmonic gold nanoparticles (AuNPs) has enabled the demonstration of plethora of reliable biosensing platforms. Here, we review the nano-engineering and biosensing applications of AuNPs based on the shape, hollow morphology, metal–dielectric, nano-assembly and heterometallic nanohybrids under optical as well as biosensing competencies. The current review emphasizes the recent past and evaluates the latest advancements in the field to comprehend the futuristic scope and perspectives of exploiting Au nano-antennas for plasmonic hotspot generation in SPCE technology.

## 1. Introduction

Different technologies are being continuously developed for achieving the successful and reliable diagnostics of human, biological and environmental health. With the advent of luminescence (fluorescence, phosphorescence, chemiluminescence) spectroscopy, numerous biosensing applications have evolved by leveraging the expertise of physicists, chemists, biologists and material scientists [1,2,3]. To further boost the performance of fluorescence-based biosensing technologies, plasmonic nanoparticles (NPs), nanohybrids and their nanocomposites with dielectric nanomaterials are extensively researched [4,5,6]. The utilization of plasmonic nanomaterials to augment the signal intensity from weakly fluorescent biomolecules and monitoring the shifts in the plasmonic resonant modes induced by several physico-chemical interactions in nano-dimensions and single-molecular limits has been explored [7,8,9,10,11]. The ability of the plasmonic nanomaterials to generate high-field intensity in their vicinity has attracted researchers from different fields [12,13,14,15,16,17,18,19]. In this regard, different shape, sizes, morphology, topography and geometrical architecture (core-shell, decorated, alloy) are explored to integrate their utility in nanophotonics, metasurfaces and miniaturized optoelectronic devices [20,21,22,23]. The excitation and emission intensities of the radiating dipoles are dramatically influenced by the high electric field profile of the plasmonic NPs, thereby resulting in significant changes in the overall quantum yield and lifetime [24,25,26]. The metal-fluorophore system effectively constitutes a resonant hybrid termed ‘plasmophore’ (plasmon + fluorophore) because emitted photons from the radiating dipoles trigger newer charge density perturbation pathways to generate plasmons that finally radiate into the far-field, carrying the attributes of both the metal and the fluorophore species [4,24]. In the process, the newer channels of radiative decay rate are realized with high photostability and dwindled lifetime [27,28,29]. This field of research has evolved into metal-enhanced fluorescence (MEF) where the conventional limitations of low sensitivity in fluorescence spectrophotometry are circumvented. A plethora of hybrid systems made up of plasmonic metal NPs and fluorescence molecules are employed based on the application of interest [30,31,32,33,34,35]. Such MEF systems are utilized for applications concerning forensics, immunoassays, single molecule detection, spectro-plasmonics and food and drug analysis, to name a few.

Surface plasmons are quasiparticles that are specially confined free electron densities (also referred to as charge densities), typically observed over the metal surfaces where the translational invariance is broken (at right angles to the geometry of the surface) [36,37,38]. The subwavelength spatial confinement and modal field of the plasmonic NPs are generated upon excitation by quantized energies (such as photons, phonons and electrons). The nanointerface at the junction of semi-infinite metal and a semi-infinite dielectric is of utmost importance as the field intensity in these regimes can be modulated upon the appropriate conditions and by the incorporation of newly designed materials (materials with positive and negative permittivities) [20,38]. Upon satisfying the phase matching condition between the incoming light radiation and the wave vector of the surface plasmons over the metal surface, the coupled wave vector referred to as surface plasmon polariton (SPPs) (where polariton is the coupled oscillation of photon and plasmon) is generated. Hence, generated SPPs inherit the properties of exponential decay into both the metal and the dielectric media in line with the solutions from Maxwell’s equations [2,38,39,40]. By and large, there are three types of plasmon resonances that are observed in the broad field of metal plasmonics, namely: (i) localized surface plasmon resonance (LSPR), (ii) delocalized surface plasmon resonance and (iii) propagating surface plasmon polaritons (SPPs) [41,42,43,44]. The LSPR signals are often observed in the case of plasmonic nanomaterials, and a simple example is the absorbance of gold (Au) NPs occurring at ~530 nm for size ~20 nm. The propagating SPPs are generated at the metal–dielectric interface and carry the attribute of propagation along the nanointerface with high sensitivity to the slight changes in the refractive index of the surrounding medium [45,46,47]. The delocalized SPR can be visualized as a partially propagating plasmonic wave that is typically observed in metal nano-assemblies where the plasmons travel within the gaps, vortices, voids and edges [48,49,50,51]. A rich spectrum plasmonics-based biosensing frameworks have utilized different combinations of the three types of plasmons for achieving high performance such as signal (in spectroscopy) or contrast (in imaging) enhancement [13,52,53,54].

Although different types of plasmonic materials have been explored globally, AuNPs are considerably distinct and stand out on account of several advantages such as (i) overarching chemical and physical stability, (ii) feasibility for bio-functionalization, (iii) copious electron density (≈5.90 × 10^16^ m^−3^), (iv) excellent photo-plasmonic response in the visible and NIR region and (v) biocompatibility, to name a few [55,56,57,58,59,60]. The SPR characteristics of AuNPs are widely used in research fields encompassing plasmon-enhanced fluorescence (PEF), heat generation, photocatalysis, energy research, non-linear optics and Raman spectroscopy to name a few [61,62,63,64,65,66]. Moreover, the surface-enhanced spectroscopic and imaging technologies have utilized AuNPs extensively in fields including but not limited to surface-enhanced Raman spectroscopy (SERS), surface-enhanced fluorescence (SEF), surface-enhanced infrared absorption (SEIRA) spectroscopy and surface plasmon-coupled emission (SPCE), to name a few [55,67,68,69,70,71]. It is worth emphasizing that the LSPR, delocalized SPR and propagating SPP characteristics of Au nanosystems have been elaborately investigated. Among such explorations, SPCE technology has evolved by adopting unique nano-engineering and biochemical functionalization routes with a paradigm shift towards translational and convergent research.

Upon interfacing the fluorescent species over the metal thin film (~50 nm) and exciting under appropriate conditions (to satisfy the phase-matching conditions), the emitted photons from the fluorescent species couple with the SPPs of metal thin film causing the coupled plasmons to radiate the fluorescence emission to the far-field and consequently carrying the characteristics of both the fluorescent species and metal thin films [46,72,73]. This is achieved by exciting the SPPs via Kretschmann configuration or by directly exciting the fluorophores as in reverse Kretschmann (RK) configuration. As the emission from the radiating dipoles are resonantly coupled to the SPPs of the metal thin film, the resulting platform is referred to as surface plasmon-coupled emission (SPCE). The SPCE signal presents several advantages over the conventional fluorescence and MEF-based signals. Due to the evanescent field coupling effect in the SPCE platform, the background signal is significantly reduced. Additionally, sharply directional and highly p-polarized emission attributes assist in the realization of more than fifty percent signal collection efficiency [4,72,74,75,76]. The amplification of the field intensity in the vicinity of the plasmonic thin films results in augmented overall fluorescence intensity. Effective nano-engineering (in the dielectric part of the metal–dielectric nanointerface) using nano-engineered materials results in the generation of nanogaps between the NPs and the metal thin film, constituting extreme field localization called plasmonic hotspots [77,78,79].

Depending on the target application, different types of metallic and dielectric thin films are incorporated to excite the plasmonic frequencies ranging from ultraviolet, visible to near and far infrared regions of the electromagnetic (EM) spectrum. Although a number of metal, dielectric, low-dimensional substrate-based nanomaterials (0D, 1D, 2D) have been explored in the SPCE platform, the unique physicochemical properties of AuNPs render them ideal candidates for SPCE studies demonstrating excellent sensitivity, reliability, reproducibility, and large linear response [1,26,53,80,81]. Although there are several reports that present the utility of plasmonic AuNPs and their variants for different types of applications, it is essential to note that a comprehensive review that presents key highlights of the utility of plasmonic AuNPs in the SPCE platform is not reported hitherto. In this background, this review aims to capture the insights obtained by studying differently engineered AuNPs in terms of shape, hollow, metal–dielectric, nano-assembly and heterometallic hybrids over the SPCE platform. The effective plasmonic nano-antennas hence realized have unparalleled ability to concentrate and modulate the electromagnetic (EM) radiation beyond the diffraction limit. These aspects are discussed presenting their utility for biosensing application despite the fact that the practical applications of SPCE go far beyond just the development of biosensing platforms.

In addition to this, the complex spatial distribution of light has been evaluated using different simulation tools such as COMSOL Multiphysics and Finite-Difference Time-Domain (FDTD). To provide a comprehensive report of the state-of-the-art research in the SPCE domain, here we streamline to discuss the key highlights of exploring AuNPs in the SPCE platform. To begin with in this review, we introduce the fundamental principles of MEF, SPCE and nano-engineering in SPCE. Following this, a few case studies are presented where the generic recipe for interfacing the plasmonic AuNPs over the SPCE platform for the development of biosensing platforms is discussed. Further, the utility of metal–dielectric hybrid interfaces is introduced with the case study of a representative example of Au–silica nanohybrids. The effective nano-engineering accomplished for dequenching the otherwise quenched signal is elaborated. Further, the effect of sharp-edged plasmonic Au is presented, highlighting the collective and coherent plasmonic coupling between the Au nanostars and plasmonic thin film. The importance of studying the hollow AuNPs are detailed with specific illustrations and a few theoretical descriptions to emphasize their utility in biosensing frameworks. Further, the readers are introduced to novel methodologies of generating heterometallic nanohybrids of Au with other plasmonically active nanomaterials. The case studies are discussed with the use of AgAu nanohybrids synthesized using frugal and sustainable nano-engineering approaches. The utility of plasmonic nano-assemblies fabricated via cryosoret nano-engineering and interfaced with SPCE platform with and without low-dimensional substrates such as graphene oxide (GO) is elaborated. We believe that this review would serve as a resourceful document for the researchers working in the domain of MEF, SPCE, nano-engineering, chemical physics and point-of-care diagnostics technologies.

## 2. Fundamentals of Nano-Engineering in SPCE Technology

In the early 2000s, the understanding of the interaction between the light emitting species such as fluorophores (also referred to as radiating dipoles) and the metallic (also referred to as plasmonic) NPs was revised significantly on account of the newer pathways established by radiating plasmon model introduced by Lakowicz and co-workers [72,73,74,82]. The readers are encouraged to refer to earlier classic reviews and research articles to understand the complete theoretical aspects from the associated electromagnetic equations and physics principles governing such interactions between metals and the radiating dipoles [2,83,84,85]. In this section, we provide physical interpretation of such interactions to facilitate easy understanding of the concepts described in the subsequent sections. The readers interested predominantly in the application aspects of the SPCE technology are encouraged to skip this section and move onto subsequent sections.

The conceptual schematic of the interaction between the radiating dipole and the plasmonic NP, metal thin film and their combination is illustrated in Figure 1. To maintain simplicity in the graphical representation, we have omitted showcasing the entire SPCE optical configuration (which is presented in earlier works as well as in Figure 2). Typically, the optical cross-section (also referred to as extinction coefficients) of the plasmonic NPs is significantly higher (~10^5^) than that of the radiating dipoles due to the strong interaction between the electric field lines and the plasmons of the metal NP [24,83]. This is often observed as the field lines curving into the plasmonic NP in the literature, a typical representation used in the lectures at the universities. Such characteristics of extraordinary optical cross-sections for field scattering are often used in biological labelling, imaging and sensing applications. Interfacing the radiating dipole with the plasmonic NP provides opportunities to modulate the Stokes shift of the fluorescence on account of the high effective extinction coefficient. On account of the enhanced localization of the electric fields of the incoming light around the plasmonic NPs, the concentrated field lines drastically enhance the excitation of the fluorophores. While this effect has been established for several years, Lakowicz et al., demonstrated another interesting phenomenon that occurs at the interface between the metal and the radiating dipole [74]. In the series of articles titled “Radiative decay Engineering—1–8”, this new aspect of light matter interactions of plasmonic NPs as well as metal thin film with the light emitting radiating dipoles have been elucidated [4,72,73,74,86,87,88]. The concept of radiating plasmon model has been introduced to explain the observations made in the metal–fluorophore resonant interactions. Briefly, the fluorophores that are excited by the incident light can in turn create plasmons by non-radiatively transferring the energy to proximal plasmonic NPs. Such a transfer of energy results in the generation of plasmons that eventually radiate to the far-field, carrying the photophysical characteristics of the fluorophores. In other words, it can be visualized as the hybrid system (of plasmonic NP and the fluorophore) itself radiating the emission rather than just viewing the fluorophore as the only emitting species. As such, coupled emission occurs extremely fast (closer to the plasmon decay rates of the metals, ~10–50 fs), the experimentally observed lifetime of the fluorophore emission decreases which re-emphasizes the fact that the plasmon is radiating the coupled photons. As the emission carries with it the properties of both the interacting species and the hybrid emitting species, it is referred to as plasmophore (plasmon + fluorophore) [4,24].

Figure 1a presents a representation of the interaction of plasmonic NP and the radiating dipole on a glass substrate. In such a scenario, the metal-enhanced fluorescence is facilitated by the property of the plasmonic NPs to concentrate the field intensity in the ambient region and, due to the enhanced scattering, the emission intensity increases with decrease in the lifetime of the emitting species. In this case, the emitted photons from the radiating dipoles strongly interact with the LSPR of the plasmonic NPs and the coupling occurring between the two assists in the observation of high fluorescence intensity. Figure 1b shows the interaction occurring between the radiating dipole and the metal thin film (~50 nm). Here, the emitted photons excite and generate the plasmons that radiate by carrying the properties of both the emission and the polarization attributes of the light (as defined by the SPR). In this case, the lifetimes and fluorescence intensity are not significantly altered, and the collection efficiency is enhanced due to the angular phenomenon emission while out-coupling through a prism in the RK configuration. As the emission is coupled to the surface plasmons of the metal thin film, this phenomenon is often referred to as surface plasmon-coupled emission (SPCE). For metallic thin films thicker than 100 nm, the decrease in the fluorescence intensity has been documented due to the lossy surface waves inducing the quenching. Here, although the excited fluorophores induce the plasmons on metal surface, the high thickness of the metal traps the plasmons (where the generated field intensity is lost as heat, Ohmic loss), thereby blocking the emissions into the far field. Hence, it is important to note that, for SPCE to occur at the metal–dielectric interface, the thickness of the film should be lower to avoid the high Ohmic loss and permit the radiating plasmons out-coupling the far field [3,89,90]. Further, the hybrid combination of the plasmonic NPs and the metal thin film results in newer modes of plasmonic hybridization depending on the properties of the plasmonic material used. For instance, the nanomaterials with sharp architectures, such as nanostars and nanorods, assist huge field enhancement due to the lightning rod effect. The incorporation of radiating dipoles in such a hybrid combination of plasmonic NPs and metal thin film result in significantly high fluorescence enhancements due to the hybrid coupling of LSPR of plasmonic NPs and the propagating SPPs of the metal thin film. In addition to this, the high signal collection efficiency of the SPCE platform in the RK configuration assists in further boosting the overall fluorescence enhancements.

This review focuses on elaborating the recent developments made in the field of SPCE where nano-engineering is carried out at the metal–dielectric interface, as shown in Figure 1c. In the past decade, the evanescent field generated at the metal–dielectric interface has been modulated with the use of different types of plasmophore systems. Different types of plasmonic nanomaterials made up of silver (Ag), gold (Au), platinum (Pt), palladium (Pd), dielectric nanomaterials made up of Nd_2_O_3_, TiC, TiN, TiCN to name a few have been explored at the metal–dielectric interface [25,53,91,92,93]. Moreover, the magnetically active nanomaterials as well as low-dimensional nanomaterials (0D, 1D, 2D) sustaining graphene Dirac fermions and solitons have been investigated [26,76,94]. The incorporation of several such nanomaterials at the SPCE interface presented newer insights into the underlying light–matter interactions from a physics perspective, where such carefully designed rational nano-engineering strategies enabled the understanding of phenomena such as Rabi splitting, plasmon–soliton coupling, Fabry–Perot photonic mode enhancement, Casimir force, ferroplasmon coupling, Fano resonance, quantum confinement and the Purcell effect [53,70,91,95,96,97]. By and large, interfacing nanomaterials at the metal thin film and studying the hybrid interface in different optical configurations such as spacer, cavity and extended cavity facilitated the engineering of hotspot intensity in the nanogap region between the nanomaterials and the metal thin film [78,94,98,99]. Consequently, the plasmophore rendered tuneable SPCE enhancements that were utilized for the development of myriad biosensing platforms. In this background, it is worth noting that the architecture of the nanomaterial constituting the plasmophore is of immense importance to comprehend the associated functionalities. Hence, there is a requirement to comprehensively review the effect of structural architecture (shape, hollow to name a few), assembly formation as well as the effect of nanohybrids including metal–dielectric and heterometallic on the performance of the resultant plasmophore towards fluorescence enhancements. In this perspective, the focus is to present the latest developments in the field of such nano-engineering at SPCE interface by considering the plasmonic AuNPs and their hybrids. To begin with, we discuss the shape effects by considering the elongated and star-shaped structures, followed by presenting hollow nanostructures and their relevance in biosensing. In the classic work titled ‘Hottest Hotspots from the Coldest Cold: Welcome to Nano 4.0′, Rai et al. demonstrated the utility of plasmonic nano-assemblies termed cryosorets to yield effective photoplasmonic hotspot engineering [23]. The highlights from incorporating the plasmonic Au with low-dimensional graphene oxide in cryosoret nano-assembly is presented in this regard. The metal–dielectric and heterometallic nano-engineering are further discussed and case studies that are relevant for the development of biosensing technologies are presented.

**Figure 2 nanomaterials-14-00111-f002:**
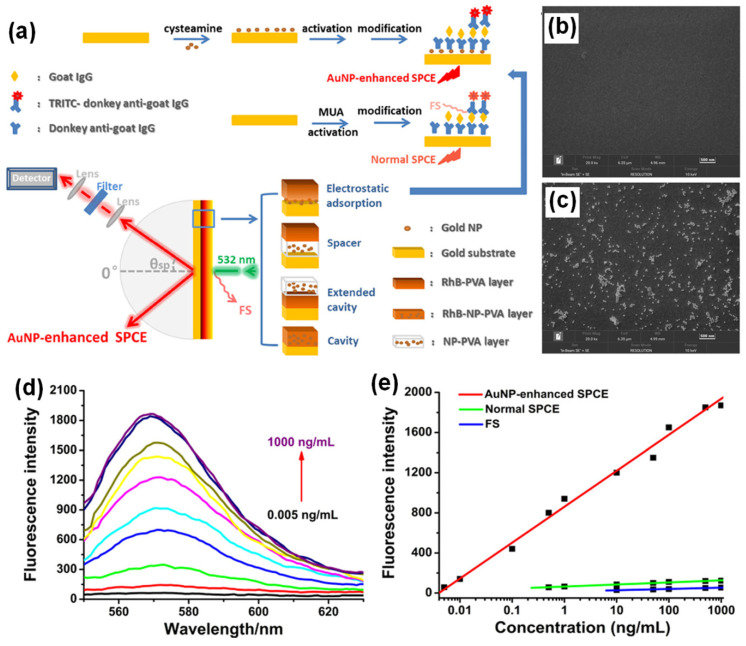
(**a**) Experimental setup for AuNPs-enhanced SPCE employing different architectures and schematic illustration of the steps for the preparation of a sandwich immunoassay. The setup is not drawn to scale. (**b**,**c**) The EDS-SEM images of AuNPs before and after loading onto the gold substrate (**d**) Fluorescence spectra for the immunosensor based on AuNPs-enhanced SPCE for different concentrations of goat IgG. (**e**) Dependence of the fluorescence intensity of AuNPs-enhanced SPCE (red), normal SPCE (green) and FS (blue) on the concentration of goat IgG. Adapted with permission from [100].

## 3. Effect of Pristine AuNPs in SPCE

Different methodologies have been adopted to interface the plasmonic materials with the light emitting radiating dipoles. In this review, we focus on the method of electrostatic adsorption and spin-coating. Here, we briefly present the advantages and importance of each of these methodologies. The process of electrostatic adsorption is a chemical process where the surface of interest is factionalized with specific chemical agents so that the desired material of interest can be electrostatically adsorbed on the surface. These methods are highly reliable as they generally give great coverage of adsorbed molecules over the plasmonic interfaces due to electrostatic interaction. However, the method of electrostatic adsorption is time-intensive and demands the development of specific surface functionalization methodologies. On the other hand, the key highlight of the spin coating method is the time-effective approach where it enables the researcher to fabricate surfaces of interest with less time. However, the demerit of using the spin coating method is the need for a clear understanding of the interaction between the radiating dipoles, polymer and the plasmonic nanomaterials. The addition of these samples to one another can be modulated to obtain the desired outcome. For instance, the direct addition of the nanomaterial to the polymer solution and further addition of the fluorescence dye solution would result in a variable fluorescence enhancement depending on the concentrations of the polymer used. If higher concentrations are used, then the plasmonic NPs can become encapsulated with a polymer layer, generating a spacer layer between the subsequently added dye solution and the nanomaterials. Hence, it is of paramount importance to consider the aspects of polymer concentration and spin coating parameters very carefully to realize effective fluorescence enhancements. Additionally, the need for the use of specific polymer solution in the case of the spin coating method is another drawback as compared to the electrostatic adsorption, as there is no need for the incorporation of polymer in the latter method.

By introducing AuNPs of different architectures on a gold substrate through electrostatic adsorption and spin-coating methods, multiple coupling structures for the SPCE system were facilitated, leading to 40- and 55-fold enhancements compared to free space (FS) emission, respectively. The diverse enhancement effect is due to novel “hot-spot” plasmonic structures, an intense EM field within the system, plasmonic properties and the process of coupling. Spin-coating based deposition of AuNPs can be used to easily build new enhancing systems with high efficiency without complex modifications, whereas the electrostatic adsorption of AuNPs provides a uniform modification, establishing highly sensitive and stable platform, which can broaden the application of SPCE in both fluorescence-based sensing and imaging.

LSPR occurs due to the energy transfer from the excited light to the collective oscillation of free electrons around the NPs, leading to a strong absorption of optical energy. The absorption peak depends on the size, shape, aggregation state, and dielectric environment. Nanoscale contacts between metal structures to form nanogap junctions are crucial plasmonic geometries. Hence, the close junctions between metallic NPs and continuous films are effective plasmon modulation systems that can induce intense interactions between localized and propagating surface plasmons to generate “hot-spots” along with enormous enhancement of the EM field under certain conditions. This phenomenon can be used to build new plasmon-based enhanced systems like SPR which has been increasingly applied in sensing and clinical detection because of its sensitivity to refractive index variations, label-free detection, and real-time monitoring [100].

A schematic diagram of the sensing procedure of goat IgG is shown in Figure 2a. AuNPs were deposited on the gold substrate by electrostatic adsorption. The carboxyl group activation, incubation, blocking buffer treatment and establishment of a sandwich structure by treatment with labelled anti-goat IgG were completed based on specific antigen–antibody interactions. In an immuno-sensor employing normal SPCE, an antibody–antigen sandwich structure was established on the modified gold substrate [100]. The RK configuration was used in SPCE platform to collect the emission after surface plasmon coupling. In the spin coating approach of introducing AuNPs on the surface of gold substrate, AuNPs and RhB were both dissolved in polyvinyl alcohol (PVA) and then spin-coated on the gold substrate in three different structures: spacer, extended cavity and cavity as shown in Figure 2a. The energy-dispersive system (EDS) spectra and SEM images of the gold substrate with and without AuNPs modification are shown in Figure 2b and 2c, respectively, to demonstrate the successful introduction of AuNPs. After the electrostatic adsorption of AuNPs on the gold substrate, the emission signals were enhanced due to the intense interaction between the fluorophore bound on antigoat IgG and the surface plasmons modulated by AuNPs modification [100]. In addition, a linear relationship (R = 0.995) was established between the AuNPs-enhanced SPCE signal and goat IgG concentration in the range of 0.005–1000 ng/mL with a detection limit of 0.005 ng/mL (Figure 2e). The linear ranges for normal SPCE and FS presented in Figure 2e, indicates that a higher detection sensitivity, a wider detection range, and a much lower detection limit could be achieved by the electrostatic adsorption of AuNPs in the SPCE system [100]. At this juncture, it is worth mentioning the insights into the specific linear relationship established between the AuNPs-enhanced SPCE signal and goat IgG concentration and its contribution to the overall performance of the immuno-sensor with the incorporation of the plasmonic AuNPs. Interfacing the plasmonic AuNPs over the metallic thin film of SPCE platform results in the coupling of propagating SPPs (of metal thin film) and the LSPR of the plasmonic AuNPs [100]. Consequently, the field intensity in the ambient region above the metal thin film is significantly enhanced due to the formation of plasmonic hotspots. Such a hybridization of modes drastically influences the emission intensity of the radiating dipoles that are used as tags for development of the biosensor. The higher concentration of goat IgG would result in the capturing of the larger number of radiating dipoles over the nano-engineered SPCE substrate [100]. Consequently, the ability to detect the goat IgG over the nano-engineered SPCE substrate is improvised due to the better sensitivity rendered by the incorporation of plasmonic AuNPs over the SPCE substrate. That is, high fluorescence enhancements in turn assist in presenting a better resolution between the adjacent concentrations in the linear range of detection. Hence, embedding AuNPs in the plasmonic structure through multiple architectures leads to a variety of interactions between surface plasmons and fluorophores, giving rise to diversified enhancement effects which can be applied to both biochemical sensing and modulated system building with high sensitivity and improved performance.

## 4. Effect of Au-Decorated SiO_2_ (AuSil), Metal–Dielectric Nanohybrids in SPCE

There has been extensive use of AgNPs in different templates as a spacer material in SPCE over several years in biochemical research for multifarious applications. Unlike silver, which suffers from quick oxidation and parasitic material Ohmic losses, gold exhibits superior chemical stability and performs excellently in visible and NIR spectral regions. However, gold has not been significantly utilized in different plasmonic nanointerfaces employed in SPCE substrate engineering due to the large unproductive quenching of commonly used fluorescent molecules such as rhodamine dyes encountered with the use of AuNPs. Au is also plagued by huge interband losses in the visible spectrum. MEF primarily depends on the distance between the fluorophores and plasmonic NPs. AuNPs significantly quench the molecular emission from fluorophores (at close distances < 5 nm). To prevent this quenching, several strategies have been employed to provide a spacer layer around the AuNPs to avoid direct contact with fluorophore [101].

Bhaskar et al., have demonstrated a rapid and facile strategy to overcome this limitation by using Au (metal) decorated-SiO_2_ (dielectric) NPs (AuSil) [101]. In this study, hybridized AuSil dequenches the otherwise quenched fluorescence emission from radiating dipoles. The comparison of transmission electron microscope (TEM) images of bare SiO_2_ NPs (Figure 3a) and synthesized AuSil hybrids (Figure 3b–d) clearly show the decoration of silica surface with gold nanospheres. The lattice fringes with d-spacing of 0.235 and 0.204 Å seen in the high-resolution transmission electron microscope (HRTEM) image of AuSil hybrid (Figure 3d) correspond to the Miller indices (111) and (200) of FCC gold. The particle size distribution histogram along with the percentage distribution for AuNPs and SiO_2_ NPs is presented in Figure 3e,f with the average size being 5.7 (±0.8) nm and 20.3 (±4.5) nm, respectively [101].

The careful choice of nanomaterials used for AuSil hybrid and the appropriate design of extended cavity nanointerface led to an 88-fold enhancement on the SPCE platform. This enhancement obtained with AuSil hybrid increased to 207-fold upon addition of 100 μM concentration of spermidine (Figure 3g), whereas decreasing the concentration of spermidine up to 10 fM allowed for the reliable and sensitive detection of this molecule [101]. The SPCE intensity spectra presenting a clear decrease in emission intensity is in consonance with enhancements (Figure 3h). The work also established a simplistic and reliable sensing platform for the detection of spermidine, a natural polyamine and a significant biomolecule at femtomolar concentration. The modulation in visible color of emissions from fluorescent molecules was captured using the smartphone camera, processed using the color grab Android app, and presented in a CIE chromaticity diagram (Figure 3i) along with their respective shade card portion on its right [101]. One can clearly notice a hypsochromic shift in the emission profile of fluorescent molecules as the concentration of spermidine is decreased. Modulation of the local density of states of fluorophores were observed by loading AuNPs on SiO_2_ NPs at different concentrations and studied in spacer, cavity and extended cavity nanointerfaces. The multi-fold hot-spots rendered by the AuSil nanohybrids assisted in the augmentation of emission radiation which was captured using a smartphone-based SPCE platform, presenting an excellent reliability and reproducibility as a detection system. Quench-free AuSil nanohybrids opens a new vista of exploration in other polaritonic systems such as photonic crystals, silicon, and graphene plasmonics.

## 5. Effect of Sharp-Edged AuNPs in SPCE

LSPR field enhancement effects of noble metallic NPs can be exploited to enhance the performance of diverse luminescent materials and devices, in which the spectral proximity plays an important role in increasing near-field enhancement-induced excitation (NFEE) and SPCE efficiencies. In their work, Wang et al., have simultaneously utilized the transversal and longitudinal SPR bands of elongated gold nanocrystals to match with the excitation and emission wavelengths of emitters, respectively, to achieve the most efficient enhancement in Polymer light-emitting diodes (PLEDs), which are the next-generation flat-panel display platforms [102]. 

Four types of gold nanomaterials as shown in Figure 4a–d (AuNPs, diameter = 20 nm), gold nanorods (AuNRs, length/width = 80/40 nm; 90/30 nm), and gold nano-bipyramids (AuNBPs, length/width = 80/40 nm) were employed as the plasmonic nanoantennas, according to spectral characteristics of the chosen emitter poly(2-methoxy-5-(2′-ethyl-hexoxy)-1,4-phenylenevinylene) (MEH-PPV). It was experimentally observed that AuNPs (20 nm), AuNRs (80/40 nm), AuNBPs (80/40 nm), and AuNRs (90/30 nm) demonstrated 1.5, 2.1, 2.3, and 1.2 times enhancement in luminance and 1.4, 1.9, 2.0, and 1.1 times enhancement in luminous efficiency in comparison with the pristine device, respectively [102]. Due to the double SPR bands of both AuNBPs (80/40 nm) and AuNRs (80/40 nm) overlapping well with the excitation and emission wavelengths of MEH-PPV, a maximum 2.2- and 2.1-times enhancement in photoluminescence intensity was observed, respectively. By controlling the aspect ratios of Gold nanorods (AuNRs) and gold nano-bipyramids (AuNBPs), tunability of their longitudinal SPR bands could be achieved. The device structure of plasmon-mediated PLEDs adopted ITO/nanometals/ZnO/MEH-PPV/MoO_3_/Ag is displayed in Figure 4e. The average distance between the emitter and plasmonic nanometals was controlled in the range 10–15 nm. A representative energy level diagram of the device is additionally depicted in Figure 4f. Because the electron mobility of MEH-PPV is about three orders of magnitude smaller than the hole mobility, the excitons derived from hole/electron recombination are mainly formed near the ZnO/MEH-PPV interface [102]. Thus, the EM field of nanometals can effectively promote the excitons transferring to emitting photons. The performance of the pristine device and hybrid ones with different morphologies and optimized amounts of Au nanocrystals are shown in Figure 4c,d.

Compared to AuNRs, AuNBPs are highly monodisperse, which suppresses the inhomogeneous spectral broadening for the plasmon peaks and the local electric field enhancements of AuBNPs are several times larger than those of AuNRs [102]. The integration of the NFEE and SPCE effects of AuNRs and AuNBPs derived from their double SPR modes, leads to simultaneous excitation and emission enhancements as made evident by steady-state and time-resolved PL measurements. This study not only systematically explores LSPR field enhancement effects of size and shape-controlled Au nanocrystals on the PLEDs’ performance but also demonstrates that AuNBPs can be applied in electroluminescence devices to enhance the emission intensity and efficiency.

Dissipative Ohmic losses due to electron scattering at the lattice of AuNPs leads to predominant non-radiative decay and quenching of fluorophores. While the incident light scatters, the lattice vibrations within the crystal lattice results in dissipative heat losses through absorption. Hence, the scattering and absorption (extinction) reduces the intensity of incident EM-field. The enhanced quenching of fluorophore with a distance < 5 nm from AuNPs is a fundamental caveat for exploiting the utility of AuNPs in SPCE. In order to address this limitation, AuNPs with nanostructures that can overcome quenching and yield multifold hot-spots with EM-field confinement need to be deployed to enhance the luminescence signal intensity [75].

The appropriate choice of metal thin films (Ag, Au, Cu and Pt) and NPs with desired shape, size, and aspect ratios in suitable nano-environment can mitigate the limitations of quenching. In comparison to Ag, Al, Cu and alkali metals, Au exhibits greater chemical stability with excellent performance in visible and near-infrared (NIR) spectral regions [75]. Apart from being biocompatible, its diverse surface functionalities have rendered Au, suitable for biomedical applications. NPs with sharp edges, crevices, tips and voids are known to entrap the incident light energy to subwavelength regions. Although the ‘lightning rod effect’ supported by nanorods exceptionally enhances the local electric-field, morphologically similar effects can be obtained from nanostars (NSs) [103].

Chandran and co-workers have demonstrated the robustness of gold nanostars (AuNSs) to simultaneously obtain template-free dequenched, enhanced (100-fold), highly polarized and directional emission in SPCE [75]. The overall dipole moment of the tip plasmons is maximized on account of coupling between the tip and core plasmons which in turn amplifies EM-field intensity in their vicinity. The sharp edges with vortices in AuNSs supporting enhanced electronic cloud perturbations that couple with nearby fluorophores, overcome the competing quenching phenomenon observed with spherical AuNPs to a large extent. Figure 5a shows SPR characteristic of AuNPs in the visible region of λ_max_: 530 nm. Figure 5b shows the SPR band of AuNSs with a λ_max_: 670 nm. The TEM image shown in Figure 5c presents gold nanospheres that are uniform in size with an average size of 16 ± 2 nm. HRTEM imaging further confirms the formation of polycrystalline AuNPs: lattice fringes observed with d-spacings of 2.36 Å and 2.05 Å correspond to the (111) and (200) Bragg planes of FCC gold. HRTEM images in Figure 5d,e also clearly reveal the star-shaped morphology of the reduced gold (AuNSs) [75]. Lattice fringes, as seen in the HRTEM image (Figure 5f) of a sharp arm of one of the AuNSs clearly reveals d-spacing corresponding to (200) Bragg planes of FCC gold (2.03 Å), and confirms the formation of AuNSs. Figure 5g shows that enhancements increased to 100, 88 and 76-fold in spacer, cavity and ext. cavity interfaces, respectively. The decline in emissions for the ext. cavity is on account of scattering of the incident beam by sharp edges and vortices of AuNSs [75]. The increase in SPCE signal intensity with the use of AuNSs in comparison to blank and FS emission is presented in Figure 5h along with corresponding angularity plot showing high directionality in Figure 5i. Electrodynamic calculations such as discrete dipole approximation (DDA) have shown that regions in plasmonic materials having sharp tips have enhancements several orders of magnitude as compared with incident field. This approach for realizing dequenched and augmented emissions using gold NPs without the additional templates will help realize novel approach for point-of-care diagnostics.

## 6. Effect of Hollow AuNPs in SPCE

The use of solid NPs in SPCE-based sensing has gradually expanded to hollow NPs owing to their numerous advantages. The LSPR peaks of hollow NPs can be easily shifted from the visible region to the near-IR region by changing the dimensions of the NP core and shell. The plasmon of the hollow NP exhibits a large spectral shift as a result of the change in the refractive index in their immediate vicinity, which consequently improves their sensing capability. The EM field of the hollow NP is strong, which is attributed to the intense plasmonic coupling between the inner and outer surfaces. The Gold Nano Shells (GNSs) (we have used GNSs here to distinguish them from gold nanostars, which were earlier abbreviated as AuNSs) have been used in several chemical, biological applications and surface plasmon based systems, because of its strong plasmonic properties, well-established capabilities of functionalization and sensitive environmental response.

Xie et al. have reported that by assembling GNSs on a gold substrate via electrostatic adsorption and subsequently applying a fluorophore layer by spin-coating, SPCE fluorescence signals exhibited 30- and 110-fold enhancements compared to those of normal SPCE and free space emissions, respectively [104]. After the junction of GNSs on a gold substrate, a “hot-spot” structure emerged between the gold substrate and GNS because of the interaction between the localized and propagating SPs. This produced a more intense EM field; specifically, the EM field around the “hot-spot” was twice as strong as that around the GNS, which was confirmed by comparing the scale bars in Figure 6a,b.

The TEM images of GNS hollow structures with a diameter of approximately 30 nm are seen in Figure 6c. An SPCE-based biosensor comprising GNSs assembled on a gold substrate could detect proteins (human IgG) via an established immuno-sandwich structure on the modified substrate surface (Figure 6d) [104]. The signal was observed to increase with increasing concentration of human IgG and the immunoassay results for both the normal SPCE and GNS enhanced SPCE systems is shown in Figure 6e. Moreover, a linear relationship (R = 0.994) was noted between the GNS-enhanced SPCE signal and the antigen concentration over a concentration range of 0.01 to 100 ng/mL with a detection limit of 0.004 ng/mL. In contrast, the linearity range for normal SPCE (Figure 6f) was only observed over a concentration range of 1 to 50 ng/mL with a detection limit of 0.2 ng/mL. Therefore, introducing GNSs provided the SPCE system with a higher detection sensitivity, a wider detection range, and a much lower detection limit [104].

Hence, it has been demonstrated that the GNS-enhanced SPCE system was suitable for biomolecule detection because of the scale match between the optimal fluorophore thickness and the biomolecule size, and thus was designed as an immunosensor to verify the feasibility of this system. This GNS-enhanced SPCE system provides several advantages. There is considerable signal enhancement due to the hollow plasmonic NPs without complex modifications or redundant, assistive materials, which greatly simplified the enhancing strategy. This method works effectively under a defined layer thickness (approximately 30 nm), which is very suitable for biomolecule detection and imaging. An appropriate GNS size helps to position the fluorophores in the effective coupling zone, which can provide enhanced signal with higher stability [104]. This strategy of introducing GNSs in SPCE has created a new fluorescence system based on a hollow plasmonic structure and provides a simple way to improve the detection sensitivity in fluorescence-based sensing and imaging platforms.

In another approach, Xie and co-workers utilized the gold nanocages (AuNCs) to generate high field confinement and fluorescence enhancement with factors over 150 and 600 compared with the normal SPCE and free space emission [105]. This is accomplished by using a thin 10 nm fluorophore layer over the SPCE substrate, thereby presenting an effective approach to dequench the otherwise quenched emission induced by resonance energy transfer (in normal SPCE) signal. The TEM characterization of the AuNCs is presented in Figure 7a along with the size distribution in Figure 7b. Recently, different simulation tools such as FDTD, COMSOL Multiphysics, DDA to name a few have been used to evaluate the EM hotspot intensity in the nanogaps generated by the plasmonic nanomaterials [105]. The FDTD simulations (based on Lumerical Solutions) were adopted to understand the near-field EM field patterns of a single AuNC with and without the gold substrate. 

From Figure 7c–f, we observe that a significant enhancement in the field intensity is observed not only on the outside of the shell, but also on the interior nano-gap based hotspot. Such coupling of fields from both the plasmonic hotspots within and outside the nanoshell is expected to yield dramatically high fluorescence enhancements due to the mixing or hybridization of modes resulting in higher local density of states [105]. The high field enhancement thus obtained was used by the authors to further utilize the same for development of multiwavelength detection system. This was accomplished by using GNSs and AuNCs that were mixed with quantum dots solution. It was observed that without the incorporation of the NPs in to the QDs solution, the multiwavelength detection system was not effective [105]. However, upon mixing the plasmonic NPs of different aspect ratios (shells and cubes) the emission intensity at all the desired wavelengths were enhanced. That is, with the incorporation of NPs the signals which were otherwise non-detectable were identified with the addition of plasmonic gold nanoparticles (Figure 7g–i). These results present the gold-based nano-engineering with hollow structures and different aspect ratios to be ideal candidates to accomplish multiwavelength simultaneous fluorescence enhancement and detection facilitating multi-analyte sensing as well as associated imaging [105].

## 7. Effect of AgAu Heterometallic Nanohybrid in SPCE

Plasmonic AuNPs have been hybridized with different nanomaterials encompassing plasmonic (such as Ag, Pt, Pd), dielectric (such as TiO_2_), low-dimensional substrates (such as carbon dots, graphene oxide), transition metal dichalcogenides to name a few to realize important functional properties, applicable in biosensing, energy harvesting, imaging, electrochemistry, water purification and diagnostics to name a few. In this section, we discuss a few case studies where the plasmonic AuNPs are hybridized with Ag plasmonic nanomaterials in different geometries and architectures and explored in SPCE platform. An outstanding example is the investigation of AuAg alloy nanoshuttle demonstrated for imaging application by Li and co-workers using the SPCE technology [106]. On account of the hybrid plasmonic coupling between the SPPs of Au substrate and the hybrid modes of AuAg nanoalloy, augmented image brightness, better signal-to-noise ratio and axial resolution were realized. The TEM images along with the elemental mapping of the AuAg nanoalloy presenting sharp tips at both the ends of the nano-rod morphology is shown in Figure 8a–e. Fundamentally, an AuAg nanoshuttle (NS) is fabricated with an Au nanorod (AuNR) as its core and Au and Ag epitaxial growth-based outer shell to generate sharp tips at both ends. The dimensions from the TEM analysis were utilized to understand the field enhancement in FDTD simulations and the lightning rod effect-based enhancement is observed in the Figure 8f as compared to lesser enhancement in bare Au substrate (Figure 8g) [106].

The performance of the AuAg shuttle towards imaging application was evaluated by dyeing the whole of Hela cells using rhodamine B molecule and observing the emission from the SPCE angle [106]. The dependence of the excitation angle as well as the polarization of the light on the intensity of the emission occurring from the labelled cells is shown in Figure 8h. The experiments were performed using the critical angle excitation mode to understand the cytoplasmic region away from the substrate, while the SPCE presented information about the membrane of the cell due to the high background rejection property of SPCE. From Figure 8i,j, one can clearly observe the effect of AuAg nanoalloys in enhancing the imaging analysis with amplified brightness and better clarity in visualization [106].

Recently, different types of plasmonic nanohybrids are being utilized to further boost the fluorescence intensity in the SPCE platform. Among them, certain frugal nano-engineering routes are being employed to generate plasmonic nanohybrids that support high field intensity. A frugal go-green bioinspired nano-engineering is a method where simple, user-friendly methodologies are incorporated to synthesize plasmonic NPs without involving high-end instruments and difficult methodologies [107,108,109,110]. In one such typical synthesis, plasmonic Ag, Au and AgAu nanohybrids were synthesized using a biomaterial, sericin which is a byproduct in the silk worm-based industry [111]. It has been reported that the hybrid AgAu presented higher fluorescence enhancements as compared to the Ag and Au based SPCE nanointerfaces. In this perspective, we have chosen to present the data pertaining to the same in Figure 9. The methodology of synthesis is shown in Figure 9a, where a simple mixture of sericin protein solution (SPS) and metal ions (Ag^+^ + Au^3+^ ion) subjected to UV irradiation generated AgAu nanohybrids without the addition of supplementary heat, reducing and capping agents. The increase in the intensity of the LSPR for different samples thus obtained and the DLS particle size distribution is shown in Figure 9b and 9c. A clear increase in the absorbance of the AgAu nanohybrids with increase in the particle size is observed [111].

Interfacing these nanohybrids in SPCE platform in different configurations (spacer, cavity and ext. cavity) yielded tunable SPCE enhancements. It is observed that the SPCE enhancements initially increased, reached a maximum and further decreased with increase in the particle size, where the 30 min UV-exposed sample yielded the highest fluorescence enhancements. The overlap of the experimentally obtained fluorescence intensity counts and the theoretically obtained dispersion diagram presents an excellent match as shown in Figure 9e. The angle that presented the highest SPCE is noted and the spectral information is shown in Figure 9f along with the corresponding p, s and FS spectra. The corresponding angularity plot is presented in Figure 9g for the same sample that yielded highest SPCE enhancements (30 min UV-exposed sample). The SEM imaging for the same sample at different magnifications is shown with the corresponding elemental mapping in Figure 9h, 9i and 9j, respectively. As observed from the SEM images, the samples displayed cubic configuration with excellent distribution of EM hotspots in the sharp edges of the generated nano-architectures [111]. The 30 min UV-exposed sample that yielded the highest SPCE enhancement was utilized for development of the sensing platform. The direct interaction of the rhodamine emitter molecule with the pharmacologically active drug molecule mefenamic acid (MA) was utilized for the indirect detection of MA using a smartphone based SPCE technology and the results are presented in Figure 9k–n. The increase in the concentration of the MA resulted in decrease in the SPCE enhancements with the effectuation of two linear responses in the detection capability of the sensor [111]. The high SPCE enhancement (more than 1300-fold) was observed on account of the sharp nano-architectures developed in AgAu nanohybrids. Importantly, the generation of the hybrid is performed using the key principles of go-green chemistry where bio-inspired approach with non-hazardous, biocompatible sericin protein (obtained from *Bombyx mori*) is used in the simple one-pot synthesis. While this case study demonstrates the utility of AgAu plasmonic nanohybrid for effective detection of a molecule of interest, the next section is dedicated to highlight the utility of plasmonic AgAu nanohybrids for detection of ions of interest by considering a case study of Zn^2+^ ions [111].

Eco-friendly methods from green nanochemistry perspective are gaining major importance in the synthesis of nanomaterials, nanohybrids and composites [37,110,112]. Rai et al. presented a sustainable chemistry approach for the bio-inspired synthesis of plasmonic Ag, Au and AgAu nanohybrids using the proteins extracted from the silkworm pupae, termed silkworm protein (SWP). The silkworm pupae contain about 40% proteins which are often cast-off as the byproduct in the silk production-based industries. Subjecting the SWP solution and the metal ion mixture to UV-irradiation for different time intervals yielded monometallic and heterometallic nanohybrids with tunable localized surface plasmon resonances (LSPR) [25]. Interfacing such plasmonic materials obtained via renewable sources (light-induced reaction) was further explored as salient candidates for fluorescence enhancements studies in SPCE technology. The high photo-plasmonic integrated hotspots generated between the plasmonic AgAu nanohybrids and the propagating SPPs of metal thin film yielded >1300-fold SPCE enhancements [25]. Different approaches are incorporated for avoiding the Ohmic loss-induced fluorescence quenching observed with the use of plasmonic AuNPs in the SPCE platform. Among them, the approach that utilizes the incorporation of Ag and Au into a single nanohybrid has presented intriguing results due to the high chemical stability of Au and high field confinement by Ag (low losses) [25]. Thus, realized fluorescence enhancements were utilized for detection of biologically and environmentally relevant Zn^2+^ ions at trace concentrations, using a simple chemical assemblage of Zn^2+^ ions-alizarin red S (ARS)–AgAu nanohybrid in SPCE platform. ARS is a fluorescent molecule that does not show high fluorescence intensity with 532 nm excitation. Hence, in order to capture observable emission, the AgAu nanohybrids are interfaced and 54-fold enhancement is achieved due to high field enhancement (Figure 10a). Upon interaction with Zn^2+^ ions the absorbance maximum of ARS shifts from ~400 nm to 530 nm, because of which considerable increase in fluorescence is observed (Figure 10a) [25]. The hybrid combination of Zn^2+^ ions-ARS–AgAu nanohybrid in SPCE platform resulted in 1000-fold fluorescence enhancements, which was further utilized for indirect sensing of Zn^2+^ ions and the results are captured in Figure 10. The decrease in the SPCE intensity with decrease in the Zn^2+^ ions concentration is shown in Figure 10b. The corresponding SPCE enhancements plotted along with the luminosity values obtained using a smartphone camera are shown in Figure 10c, along with the associated shade cards in Figure 10d. The results from the selectivity and the spiking studies are shown in Figure 10e–g. It is observed that the developed biosensor displayed a high sensitivity, selectivity and reproducibility towards the detection of Zn^2+^ ions, using cost-effective smartphone technology amenable for bottom of the pyramid [25]. Such explorations where generally discarded biocompatible byproducts are used for synthesis of nanomaterials can be further extrapolated to study the utility of other waste byproducts from industries including paper, sugar, electronics and aquaculture to name a few towards different transdisciplinary applications. Moreover, these research works opens possibilities to investigate the effect of auxiliary experimental conditions such as pH, surfactant concentration, temperature and analytes (ions and molecules) for the controlled and directed growth of plasmonic nanomaterials [25].

## 8. Effect of Au–Graphene Oxide Nano-Assembly in SPCE

Rao and co-workers in their pioneering works in the SPCE substrate development investigated the relevance of low-dimensional materials such as GO and MoS_2_/WS_2_/BN for the generation of unaccustomed photo-plasmonic hotspots [91,93]. Further, Yao-Qun Li and co-workers demonstrated the significance of GO-based π-plasmons for establishment of novel biosensing frameworks in SPCE technology [113]. Following this Ramamurthy and co-workers demonstrated the utility of hybrid combination of low-dimensional GO and metallic (Ag nanowires), dielectric (TiO_2_), magnetic (Nd-Ag nanohybrids), presenting interesting insights from plasmonics perspective. Recently, Sundaresan and co-workers demonstrated an interesting approach to dequench the molecular fluorescence from novel radiating dipoles in the proximity of plasmonic AuNPs by engineering them in the nano-assembly architecture with low-dimensional graphene oxide (GO) interface [49]. A benzoxazolium-based fluorescent molecule, (E)-2-(4-(dimethylamino)styryl)-3-methylbenzo[d]oxazol-3-ium iodide (DSBO), that presents an emission maximum at ~550 nm (under 532 nm excitation) was synthesized to selectively detect the cyanide (CN–) ions in water samples. In the vicinity of plasmonic AuNPs the emission from the dye molecules would normally be quenched as the plasmonic AuNPs have high absorption co-efficient at ~550 nm (which is the emission maximum of the radiating dipoles) [49]. In order to circumvent this caveat, cryosoret nano-engineering (CSNE) technique was used for efficient generation of Au nano-assembled in the presence of GO, gold-graphene oxide-cryosoret (AuGOCS) [23,49]. Such hybrid combination of metal-π plasmon resonant coupling assists in the generation of nanovoids (between the NPs of the nano-assembly) and nanocavities (between the NPs of the nano-assembly and the plasmonic thin film) assisting delocalized Bragg and localized Mie plasmons. The interfacial coupling components used for the generation of hottest hotspots (regions of extreme field confinement and enhancement) are shown in Figure 11a. Multiple TEM images of the AuGO cryosorets are shown in Figure 11b along with the HRTEM image in Figure 11c. The SPCE enhancements recorded with different combination of samples are shown in Figure 11d. The AuGOCS + GO interface yielded the highest dequenched and augmented fluorescence enhancements. Here, the additional GO is attributed to the monolayer of GO performed over the SPCE substrate (Ag thin film) prior to interfacing the AuGOCS. The SPCE, p, s and FS spectra for the samples yielding highest SPCE enhancements is shown in Figure 11e, along with the experimentally obtained directionality plot in Figure 11f.

The results from the SPCE based sensing analysis are shown in Figure 11g–k. A direct addition of CN^−^ ions to the DSBO fluorescent molecules resulted in strong quenching due to the chemical interactions between the molecules and the ions. As shown in Figure 11g conceptual schematic, this quenched emission can be dequenched with the use of AuGOCS with concomitant realization of high fluorescence enhancements. This high SPCE enhancement was further utilized for the indirect detection of CN^−^ ions in water samples. The SPCE enhancements (as obtained using the Ocean Optics spectrophotometer) are plotted along with the luminosity values obtained using the smartphone-based detection system [49]. The developed dequenching (turn-on) of the quenched (turn-off) fluorescent signal successfully demonstrated attomolar limit of detection of 10 aM of CN^−^ ions with high linearity (R^2^ = 0.996). Following the pioneering work in cryosorets by Rai et al. the classic work titled ‘Hottest Hotspots from the Coldest Cold: Welcome to Nano 4.0′ is envisaged to be of immense utility for the generation of novel strategies for sensitive and selective determination of environmentally and biologically relevant analytes at extremely low concentrations for use in point-of-care diagnostics [23,49].

## 9. Limitations and Challenges

No doubt, plasmonics and photonics offers the decisive spatial and temporal sovereignty on account of the capability of nanostructures to concentrate electromagnetic energy into nanoscale volumes [2,114,115,116]. A flurry of promising experiments using different types of AuNPs in various photo-plasmonic platform demonstrate a transformative impact on the way researchers engineer, manipulate, enhance and monitor the performance of biosensing approaches [116,117,118,119,120,121,122,123,124]. Here, we discuss the major challenges, and elucidate the future scope and opportunities. To begin with, as we observed in this review, there are several means to synthesize plasmonic AuNPs, and depending on the application of interest a unique approach is implemented. A typical synthesis of plasmonic AuNPs demands the use of toxic chemicals, especially when a specific shape or size is demanded. Hence, there is a trade-off in disregarding the use of hazardous chemical entities (molecules, ions and polymers) as well as solvents. The synthetically obtainable capping and reducing agents are often toxic and detrimental to the living systems (both terrestrial and aquatic). Moreover, the physical methods of nanosynthesis demands additional use of energy and resources in the form of supplied temperature, pressure and sophisticated setups. In this background, the motivation for employing the bio-inspired routes for nanosynthesis has evolved where the principles of green chemistry are taken into consideration. Although frugal approaches utilizing UV-light-induced synthesis and temperature-based cryosoret nano-engineering presented precise nanofabrication approaches, there are challenges that need to be addressed [23]. In the case of UV-light based synthesis approaches, it is important to modulate the process conditions to achieve high monodispersity in nanofabrication. On the other hand, the cryosoret nano-engineering method needs to incorporate steps for realization of directed and hierarchical self-assembly of plasmonic AuNPs. This is because the current approach renders nano-assembly highly anisotropic. In this perspective, altering the conditions of the experimental workflow by incorporation of changing pH and variable concentrations of bioinspired surfactants would enable better control over the monodispersity as well as the directed self-assembly of NPs.

The exploration of fluorescent moieties that respond with different wavelength regions (such as UV, vis, near and far NIR) demand clear understanding of the scattering and absorption profiles of the synthesized nanosystems. This is extraordinarily relevant in the case of MEF and SPCE because the radiative decay rate of the fluorescent entities (such as organic molecules, fluorescent polymers, quantum dots and nanodiamonds) depend on the scattering cross-section of the plasmonic NPs in the close proximity of the light-emitting species. Appropriate simulation and experimental tools can be utilized to decouple the absorption and scattering components of the overall extinction spectra (which is generally recorded in the absorbance spectrophotometers). This would enable the successful identification and implementation of most suitable fluorescent entities (where the emission wavelength of fluorescent species can be matched with the scattering wavelength of the nanosystems), especially in scenarios demanding effective Förster resonance energy transfer (FRET) and related phenomena on account of a better understanding of the absorption and scattering profiles of the nanosystems.

Another major challenge that hampers the complete understanding of the processes is the negligible information available on the intrinsic physicochemical properties from a materials chemistry standpoint. The AuNPs, their hybrids with other plasmonically active materials such as Au, as well as their nano-assemblies with metal, dielectric and metal–dielectric hybrids would generate variable real and imaginary components of permittivity, which is essential to understand the regions of optical loss. Additionally, it is important to characterize the overall refractive indices of the composites and the nanohybrids to enable a clear understanding of the contributing factors towards MEF and SPCE. This is because the high-refractive-index nanohybrids not only respond to the electric field component of the EM radiation, but also resonate with the magnetic field component, hence generating magnetic hotspots. Therefore, experimentation and characterization using Electron Energy Loss Spectroscopy (EELS) and associated technologies need to be performed to understand these properties in terms of the dielectric properties of the materials. Further, advanced simulation tools might be necessary to de-couple the contribution of plasmonic AuNPs and the metal thin film towards the overall fluorescence enhancements in SPCE platform. From the photo-plasmonics standpoint, it is important to incorporate the laser irradiation frequency as close as possible to the plasma frequency of the metal considered. Plasmonic films or metals are transparent to excitation frequencies which are larger than the plasma frequency of the metal. Further, the laser irradiation should also be as high as possible in terms of frequency so as to dwindle the influence of imaginary component of the dielectric constant. In light of these considerations, it is worth noting that extensive material characterization of the synthesized plasmonic nanomaterial would enable researchers to streamline the demands of the project considering the parameters including laser wavelengths, filters, polarizers and detection systems.

## 10. Futuristic Scope and Perspectives

Plasmonics and fluorescence-based techniques (spectroscopy and imaging) are natural partners, given that the field of plasmonics renders simple and reliable means to realize point-of-care diagnostic technologies with qualitative and quantitative analyte determination. This is achieved by leveraging the intrinsic potential of plasmonic nano-entities to concentrate far-field EM radiation to sub-diffraction-limited dimensions. The recent explorations incorporating AuNPs in the SPCE platform and associated technical breakthroughs in the characterization and development of biosensing frameworks have gathered significant interest from researchers working in scientific disciplines such as physics, chemistry and life sciences.

Although there are substantial reports with the use of plasmonic AuNPs in the SPCE platform, their utility is often concealed by the so-called zone of inactivity in Au substrates where at distances less than 5 nm, the non-radiative decay channels support higher-order plasmonic modes thereby resulting in unavoidable quenching phenomena. Different approaches have been developed to circumvent this limitation: (i) plasmon intermixing in heterometallic nanohybrids in photo-plasmonic inter-cavities (between the nanohybrid and metal thin film) to facilitate dequenching of the quenched signal, (ii) core–shell metal–dielectric nanohybrids at advanced functional interfaces, (iii) metal–dielectric decorated nano-designs, (iv) nanomaterials presenting sharp nano-architectures with protrusions (nanotips) supporting hybrid tip-core plasmon coupling, (v) void or intra-cavity (within the NP framework) plasmons generated with nanoshells and nanocages [55,75,97,100,101]. In this review, results from several of these above-mentioned situations are deliberated by providing interesting illustrations and inferences that are of immediate relevance to the photo-plasmonics, biosensing, fluorescence spectroscopy and internet-of-things. The capabilities of nano-engineering can be used to generate plethora of nanohybrids using plasmonic AuNPs and the glimpse of the futuristic scope in this perspective is shown in Figure 12.

From the futuristic perspective, it is important to note that the amalgamation of fluorescence spectroscopy and plasmonics assists in augmented performance of two technologies that are parallelly growing in research interest, namely spectroscopy and imaging. While the biosensing frameworks discussed in this review focuses on the spectroscopic signals and associated outcomes (such as chromaticity and luminosity) the imaging aspects are underexplored, especially in the broad domain of super-resolution, where the plasmonic AuNPs have been utilized as excellent alternatives for organic fluorescent moieties and quantum dots to achieve image contrast. The plasmonic AuNPs in SPCE frameworks can be further explored for such purposes where the large optical cross-sections and high photostability accompanied with negligible blinking effects can be leveraged to augment the performance of super-resolution microscopy. In such explorations, the structure–activity relationships (SAR) can be evaluated for plasmonic AuNPs in SPCE technology. It is informative to mention the importance of structure–property relationship with respect to designing nanomaterials with desired functionality [125,126,127]. Significant interest has been devoted in the controlled synthesis of nanomaterials where a single particle parameter such as shape, for instance, is controllably altered by fixing other parameters such as size, stabilizing agent, pH, to name a few. Additionally, variation in the properties of the capping agents (tannic acid, polyethylene glycol (PEG), ethylenediaminetetraacetic acid (EDTA), polyvinyl pyrrolidone (PVP) and polyvinyl alcohol (PVA)) drastically influence the surface-active properties and stability of the nanomaterials [128,129]. For instance, bulky capping agents would assist in the generation of a spacer nano-layer coating over the nanomaterial, that hinders surface induced quenching effects in plasmonics based biosensing applications [130,131,132,133]. In this regard, challenges pertaining to nano-synthesis and their functionality are being considered in recent works to establish a comprehensive understanding of the performance of the device.

On the other hand, the biosensing frameworks working on the principle of spectroscopy rely on the two fundamental effects, namely, (i) the plasmon-enhanced signal from weakly fluorescent biomolecules and (ii) comprehending the shift in the resonance of the NP plasmon-molecule interfacial interactions. Although different types of fluorescent moieties are routinely investigated in the domain of biophysics and biosensing, the performance of the associated sensing technologies are limited to the use of fluorescent moieties that present large quantum yield (more than 0.1). The procedure that is often employed for feebly or negligibly emitting biomolecular structures is highly dependent on labelling methodologies by incorporating an external high-quantum-yield label. This is a major drawback of existing biosensing approaches because, although the technique is reasonable for proof-of-concept demonstrations, their feasibility and reliability in applications related to biosensing in bodily fluids is uncommon. In this context, there is a driving need for exploring platforms that aid successful implementation of label-free biosensors without the requirement of high-quantum-yield labels. In this background, the SPCE platform in combination with AuNPs and their variants are expected to be of immense utility with the potential to facilitate single molecular level recognition.

The plasmon-controlled fluorescence-based applications such as SPCE are highly compatible with the development of biosensing platforms where concentrated biomolecular species can exist as in bodily fluids. For instance, blood serum consists of 60–80 g per liter of total protein content encompassing albumin and globular moieties. It is observed that an optical biosensor desired for the detection of analytes in such an environment needs to respond to less probe volumes because the clinically active biomarkers occur in nanomolar to micromolar concentrations. The far-field microscopic techniques suffer in this regard because the probe volume is ~1 fL, thereby constraining the detection limit to nanomolar range. On the other hand, the probe volume of plasmonically active species depends on the magnitude of local field enhancement. As the field intensity extends to cover a few zeptoliters, the sensitivity of the desired biosensor can be further enhanced, enabling single-molecule detection at relatively high concentrations (microliters). In this regard, different approaches have been explored to further enhance the local EM field intensity on the SPCE platform. While the use of plasmonic AuNPs facilitates this goal, recently, the use of different types of semiconductors, metals, low-dimensional substrates and two-dimensional (2D) nanomaterials (graphene, reduced graphene oxide, graphene oxide, hexagonal boron nitride, pnictogens, MXenes, metal oxides and non-metals) displaying distinct plasmonic effects have been investigated. Additionally, several opportunities for future research with the AuNPs in SPCE platform includes utilizing these nanomaterials to understand the nanomaterial–molecular interaction from both physical and chemical sciences perspective. In order to accomplish this objective, the coupling of electron charge densities to lattice vibrations (called as phonons) needs to be evaluated. The comprehensive understanding of such plasmon heating phenomena (in space and time), enables chemists to study out-of-equilibrium reactions at the plasmonic interfaces. The plasmonic AuNPs along with their hybrid combinations with low-dimensional materials are expected to be of use in SPCE technology for monitoring vaporizing chemical species, plasmon-enhanced chemical vapor deposition, thermally stimulated catalysis (via hot electrons), photocatalysis and photoelectrocatalysis [134,135,136,137,138,139,140,141,142].

The permutations and combinations of different size, shape, assembly, hollow structures and heterometallic nanohybrids of AuNPs and their interfacial interactions in the SPCE platform is projected to engender unique light–matter interactions with newer insights in the broad domain of photo-plasmonics. Recently, the anisotropic plasmonic nanomaterials and the associated metasurfaces has gathered tremendous interest in the vast field of nanophotonics on account of the programmable high-quality (high-Q) optical resonances. Such optical interfaces are used for augmenting the directional emission from the semiconductor quantum dots on account of practically attainable giant Purcel factors [143,144]. Moreover, the incorporation of different types of several two-dimensional materials (including WS_2_, MoSe_2_, MoS_2_, WSe_2_ and TaS_2_) and semi-metals (TaIrTe_4_ and PtTe_2_) in the SPCE platform with and without different low- and high-refractive-index nanomaterial-based nano-assemblies with AuNPs would result in uncommon optoelectronic applications [145,146,147,148,149]. However, it is important to utilize the adequate comprehension of the physicochemical attributes of these materials to arrive at a logical conclusion prior to exploring the associated synthesis protocols. It is extremely important to recognize the pitfalls of the discipline-driven investigations from the bio-inspired go-green approach and thus, it is not suitable to indiscriminately inspect the plasmonic nanohybrids without a theoretical and experimental premise. Although the simulation-based investigations provide suggestions for appropriate combination of nanomaterials based on desired application (such as biosensing photovoltaics, photo-diodes, photocatalysis, photonic lasers, etc.), the associated computational techniques require several approximations and simplifications to resemble the laboratory-based realistic outcomes. Hence, due to lack of the ability to entirely mimic the laboratory-based experimentations, the obtained results are said to be semi-quantitatively effective, therefore demanding relevant troubleshooting steps to match the experimental and computer-based results to some extent. In this regard, it is practically relevant to utilize the fortunes rendered by the advanced tools such as artificial intelligence (AI), neural network and related machine learning tools to streamline the procedures and accurately envisage the properties of the different combinations of materials (metals, dielectrics and semiconductors) at nano-regime with plasmonic AuNPs as a primary substrate material. Such a comprehensive analysis would provide appropriate integration of the optical, mechanical, chemical, and other material information of individual components from the existing vast literature with inferable characteristics from well-trained AI platforms, hence aiding in justifying the efforts attempted in a real laboratory setup. The appropriate algorithms would assist in utilizing the massive information available about the biocompatible materials such as proteins and biopolymers for screening possible sustainable chemical synthesis routes towards achieving AuNP-based photo-plasmonic hybrids with tunable functionalities and geometries. Such integration presents several advantages from the perspective of developing a suitable biosensing platform based on SPCE technology. For instance, the utility of cloud computing, data processing, storage and virtual feedback from experts can be exploited by the health practitioners to provide real-time updates on the health status of the biological (humans, animals, fishes and birds) as well as environmental (soil, water and air) aspects after performing on-site analysis using smartphone-based detection system. Such platforms would render the technology amenable for underdeveloped and developing countries, especially in resource-limited settings and we believe that this is the future of biomedical healthcare with adequate goals to cater to the guidelines and expectations of World Health Organization (WHO) that necessitates a point-of-care diagnostic technology to be affordable, sensitive, specific, user-friendly, rapid, equipment-free and easily deliverable to the end user (abbreviated as “ASSURED”).

## 11. Conclusions

Although numerous chemical and biological sensing devices are being developed for large-scale applications, there is still room for improvement when it comes to the sensitivity of the device. Surface plasmon-coupled emission (SPCE) technology has emerged as a versatile tool to accomplish this objective as it provides augmented sensitivity as compared to other related optical sensors. Among the several plasmonic nanomaterials explored in the SPCE platform, AuNPs and their modifications have attracted the scientific community encompassing physicists, chemists, biologists and nano-engineers. Here, we present a comprehensive review of the latest developments in this direction by considering different case studies. Nano-engineering and biosensing applications of AuNPs based on the shape, hollow morphology, metal–dielectric, nano-assembly and heterometallic nanohybrids are considered in this perspective. We believe that the highlights, perspectives and future scope presented in this review would be beneficial to the researchers working in the broad domain of photo-plasmonics in general, and SPCE in particular.

## Figures and Tables

**Figure 1 nanomaterials-14-00111-f001:**
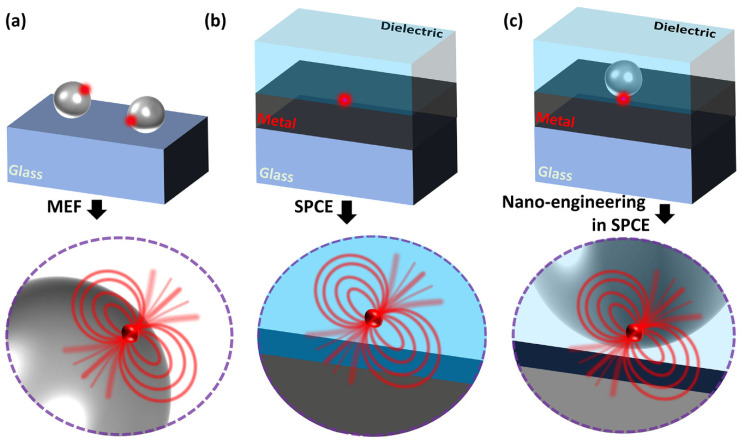
Conceptual schematic showing the interaction of the radiating dipole with the (**a**) plasmonic NP, (**b**) metal thin film in SPCE configuration, (**c**) metal thin film and a plasmonic NP in SPCE configuration.

**Figure 3 nanomaterials-14-00111-f003:**
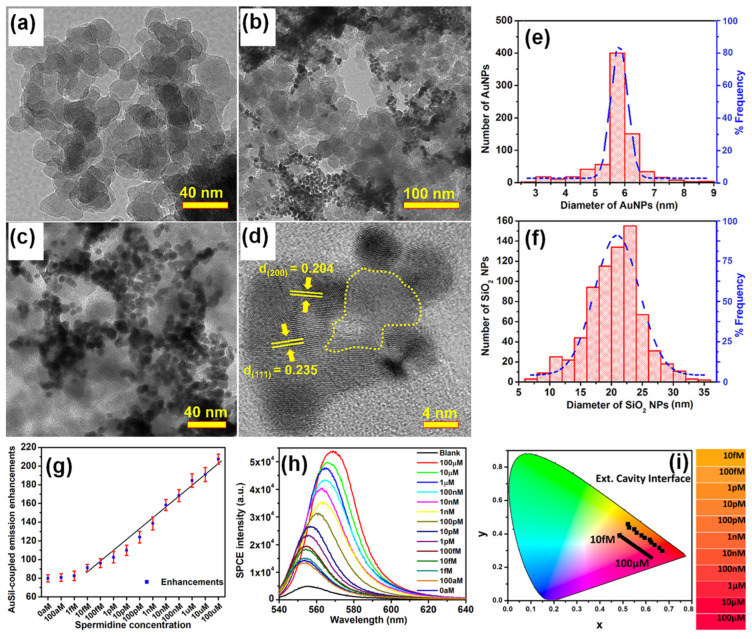
(**a**) TEM image of SiO_2_ NPs. (**b**–**d**) TEM images of synthesized Au-decorated SiO_2_ (AuSil) nanohybrids at different magnifications. In HRTEM image (**d**) yellow dotted curve indicates amorphous SiO_2_ decorated with crystalline AuNPs (lattice fringes are shown with corresponding Miller indices for gold). (**e**,**f**) Particle size distribution histogram of AuNPs and SiO_2_ NPs shown along with percentage frequency [right y-axis] obtained from TEM analysis. (**g**) Plasmon-coupled emission enhancements for increasing concentrations of spermidine taken along with AuSil hybrids in ext. cavity interface. (**h**) SPCE spectra for corresponding emission enhancements observed in (**g**). (**i**) CIE chromaticity plot and corresponding shade card indicating color change in emission pattern consistent with results in (**h**). Adapted with permission from [101].

**Figure 4 nanomaterials-14-00111-f004:**
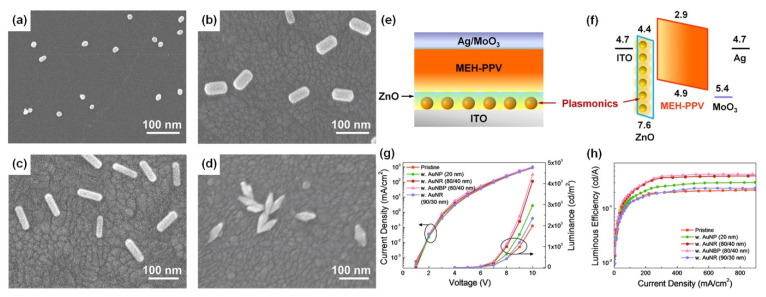
SEM images of (**a**) AuNPs (20 nm), (**b**,**c**) AuNRs (80/40 nm, 90/30 nm), and (**d**) AuNBPs (80/40 nm) deposited on the ITO substrate by the electrostatic adsorption, (**e**) schematic layout of plasmon-mediated PLEDs. (**f**) Energy level diagrams of ITO, ZnO, MEH-PPV, MoO_3_, and Ag. PLED characteristics of (**g**) current density vs applied voltage (J–V) and luminance vs applied voltage (L–V), and (**h**) luminous efficiency vs. current density (LE–J) curves, including pristine and hybrid devices with different morphologies and optimized amounts of Au nanocrystals. Adapted with permission from [102].

**Figure 5 nanomaterials-14-00111-f005:**
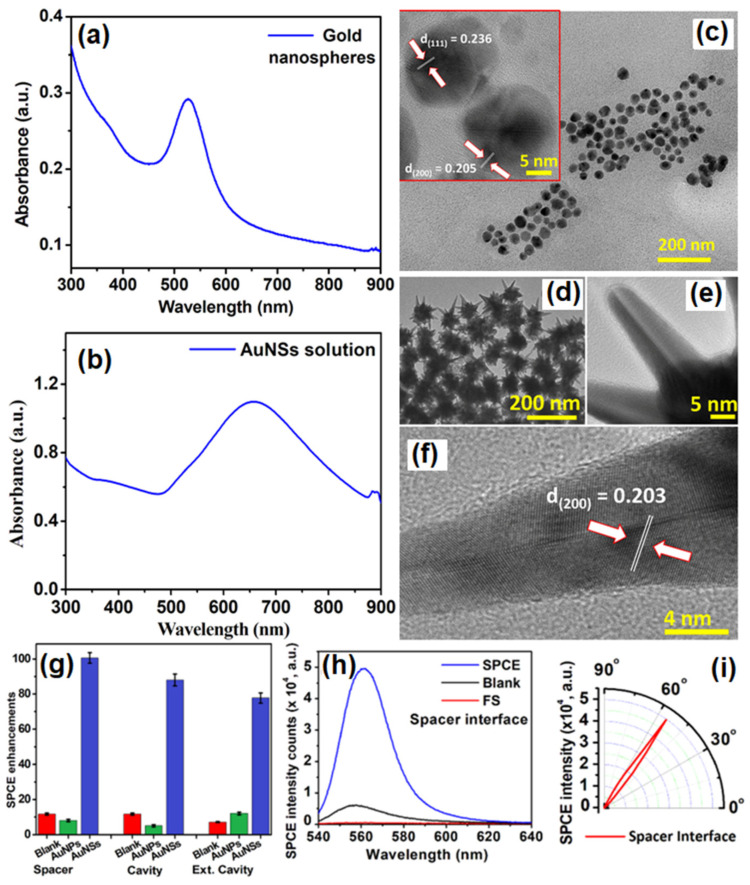
(**a**) UV-vis absorbance profile of AuNPs (**b**) UV-Vis absorbance profile of AuNSs. (**c**) TEM image of synthesized AuNPs and HRTEM image of AuNPs indicating the lattice fringes characteristic to gold (inset). (**d**,**e**): TEM images of synthesized AuNSs. (**f**) HRTEM image of AuNSs indicating the lattice fringes along with d-spacing and Miller indices characteristic of gold. (**g**) Comparative assessment of SPCE emission enhancements for AuNPs and AuNSs along with respective blanks in spacer, cavity and ext. cavity interfaces. (**h**) SPCE and FS emission intensity obtained with AuNSs along with blank in spacer interface. (**i**) Angularity plot for SPCE with AuNSs in spacer interface. (**h**,**i**) are chosen for representation because spacer interface presented maximum emission enhancements. Adapted with permission from [75].

**Figure 6 nanomaterials-14-00111-f006:**
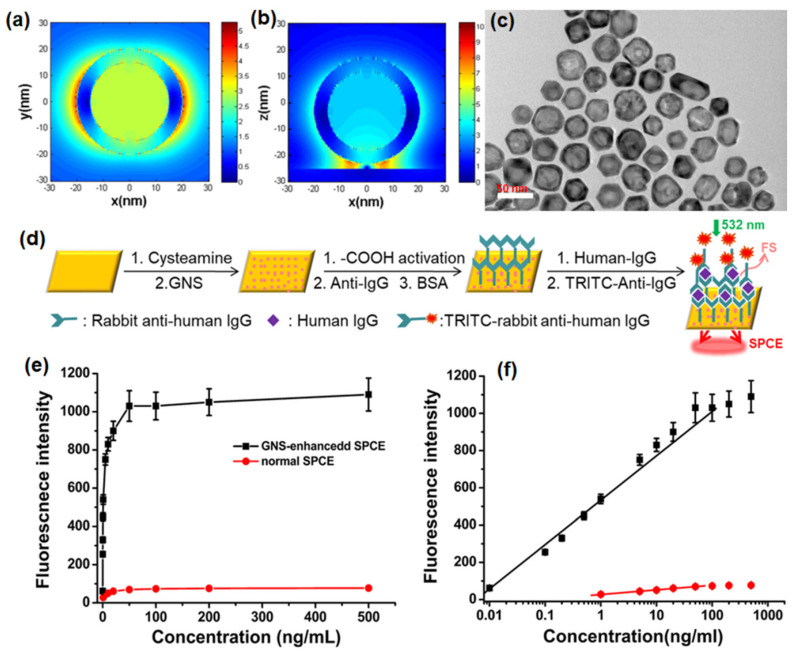
Finite-difference time-domain (FDTD) simulations for the near field region of (**a**) a single GNS (diameter: 40 nm; shell thickness: 5 nm) and (**b**) a gold substrate modified with GNS at 532 nm (diameter: 40 nm; shell thickness: 5 nm; distance between GNS and gold substrate: 2 nm; thickness of gold substrate: 50 nm). (**c**) TEM image of GNS displayed the desired hollow structures coated with a thick gold layer. (**d**) The sensing schematic diagram (this diagram is not to scale). (**e**) Fluorescence intensities for immunosensors based on GNS-enhanced SPCE (black) and normal SPCE (red). (**f**) Dependence of the fluorescence intensity of GNS-enhanced SPCE (black) and normal SPCE (red) on the concentration of human IgG. The fluorophores employed were TRITC. Adapted with permission from [104].

**Figure 7 nanomaterials-14-00111-f007:**
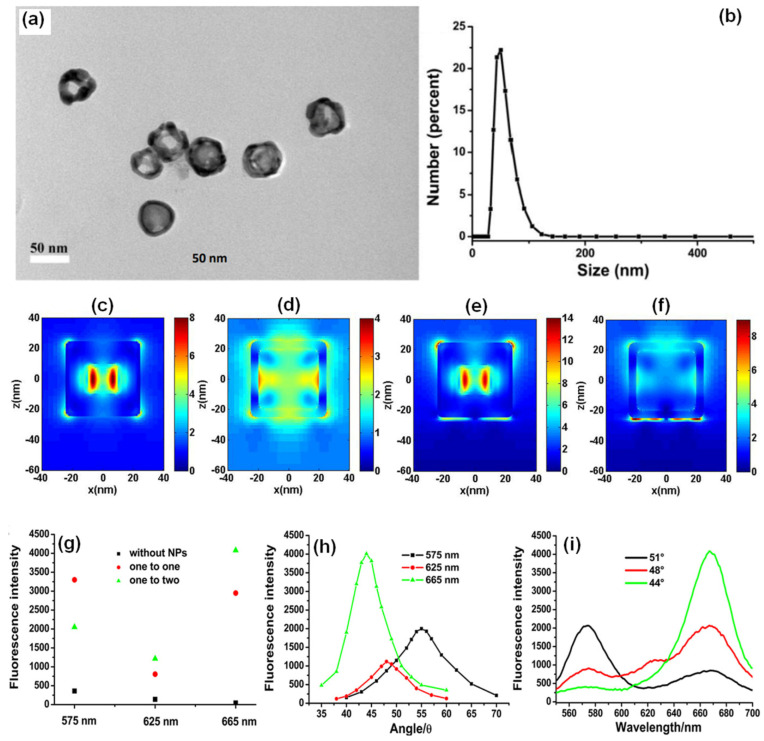
(**a**) The TEM images and (**b**) the DLS size distribution of AuNCs. FDTD simulation results of the near-field of a single AuNC and gold substrate with the AuNC at 532 nm (**c**,**e**): the outside surface; (**d**,**f**): the inside of the AuNC). Length of AuNC: 50 nm; length of hole: 20 nm; shell thickness: 5 nm; distance between AuNC and gold substrate: 2 nm; thickness of gold substrate: 50 nm. (**g**) SPCE signals of different wavelengths for different mixed ratios, (**h**) angle distributions for different emission wavelengths, and (**i**) the fluorescence spectra obtained from different optimal emission angles for different wavelengths (mixed ratio was one to two for GNSs to AuNCs). Adapted with permission from [105].

**Figure 8 nanomaterials-14-00111-f008:**
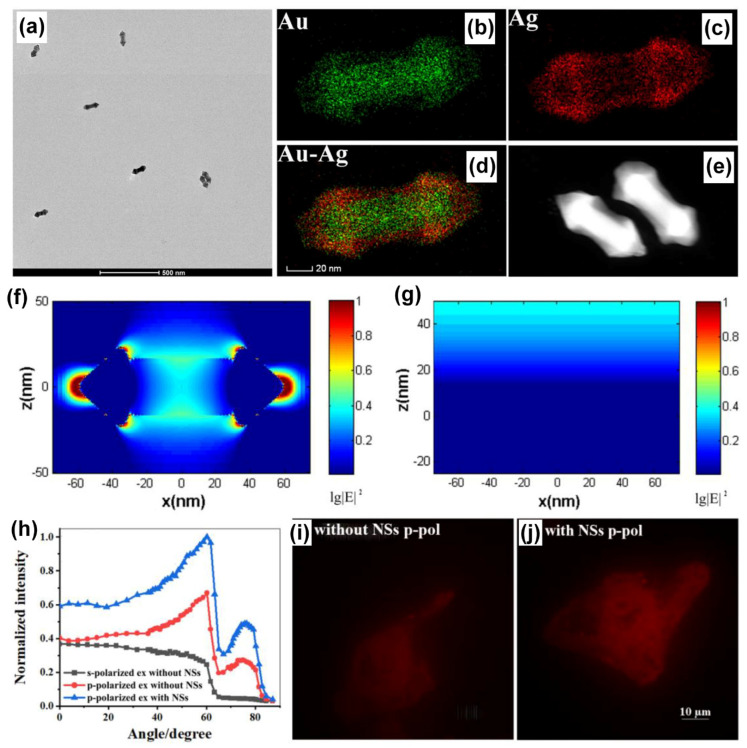
(**a**) TEM image and (**b**–**e**) mapping analysis for AuAg NSs. FDTD simulations results of the near field at 561 nm for (**f**) a single AuAg NS and (**g**) bare Au substrate. (**h**) The relationship between the normalized intensity of RhB-dyed cell images and the excitation angle. The RhB-dyed cell images from SPR excitation (**i**) without and (**j**) with AuAg NSs modification. Adapted with permission from [106].

**Figure 9 nanomaterials-14-00111-f009:**
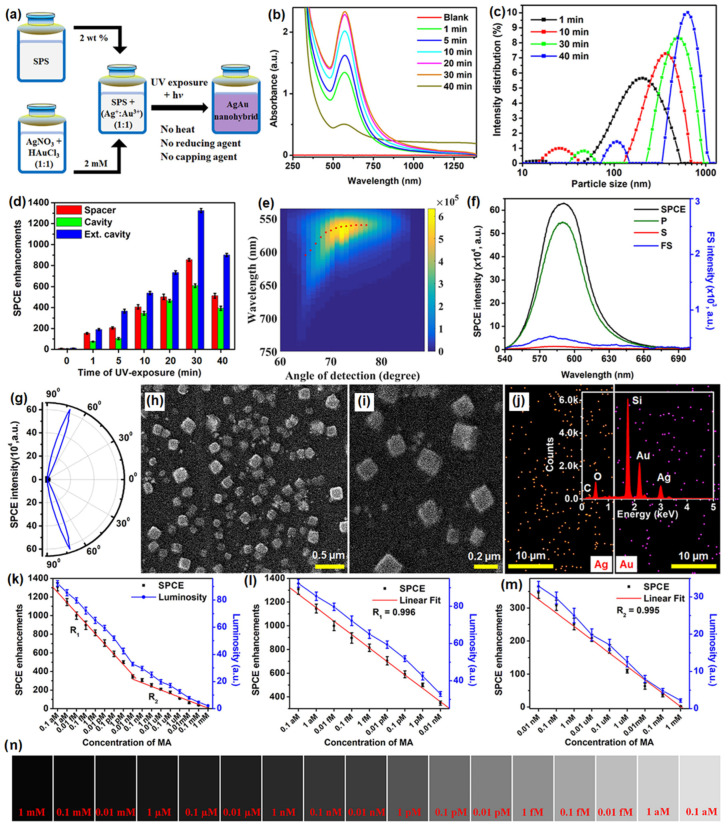
(**a**) Schematic representation of AgAu nanohybrid synthesis using sericin protein solution (SPS). (**b**) UV–vis–NIR absorbance and (**c**) DLS particle size distribution of AgAu nanohybrids obtained by exposing SPS and Ag^+^ + Au^3+^ ion mixture for different time intervals. (**d**) SPCE enhancements obtained for differently UV-exposed samples while being studied in spacer, cavity and ext. cavity nanointerfaces. (**e**) Overlap of the experimentally (shaded region) and theoretically (dots) obtained dispersion data; (**f**) SPCE, FS, p, and s emission spectra; and (**g**) sharply directional angularity plots for the 30 min UV-exposed sample. (**h**,**i**) SEM images of the 30 min UV-exposed samples at different magnifications, respectively. (**j**) Elemental mapping obtained from EDAX for Ag–Au nanohybrids along with the insets showing the chemical mapping. (**k**–**m**) SPCE enhancements and luminosity values obtained for different concentrations of MA. (**n**) Shade card of luminosity values. Adapted with permission from [111].

**Figure 10 nanomaterials-14-00111-f010:**
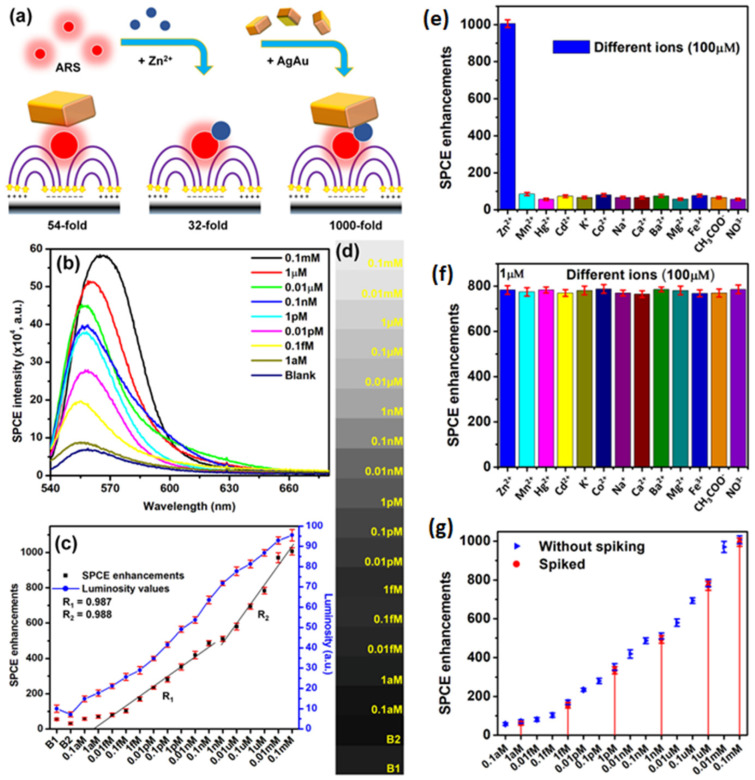
(**a**) Conceptual schematic representation of various combinations: ARS with AgAu nanohybrids (with 54-fold), ARS–Zn^2+^ ion complex (with 32-fold), and ARS–Zn^2+^ ion complex with AgAu nanohybrids in the ext. cavity nanointerface (with 231-fold enhancement). ARS is shown as red fluorescent molecules and AgAu NPs as cuboidal nanostructures in line with SEM images. (**b**) SPCE spectra for Zn^2+^ ion concentrations added to the ensemble. (**c**) Overlap of SPCE enhancements (left *y*-axis) obtained from an Ocean Optics detector and the luminosity values (right *y*-axis) extracted from the mobile phone-based detector. (**d**) Gray-scale shade cards corresponding to the luminosity values. (**e**) SPCE enhancements for different ions at 100 μM concentration. These ions were taken in the ext. cavity interface as part of the dye layer. (**f**) SPCE enhancements obtained for all the interfering ions under study (100 μM) taken in 100 times the concentration of Zn^2+^ (1 μM) for understanding the interference from other ions. (**g**) Overlap of emission enhancements obtained without spiking (Figure 10c) and from spiking experiments (red color dots shown with red lines connecting them to the corresponding Zn^2+^ ion concentration in the x-axis for ease of understanding). For spiking studies, 100 μM, 1 μM, 1 nM, 1 pM, 1 fM and 1 aM concentrations were chosen. Adapted with permission from [25].

**Figure 11 nanomaterials-14-00111-f011:**
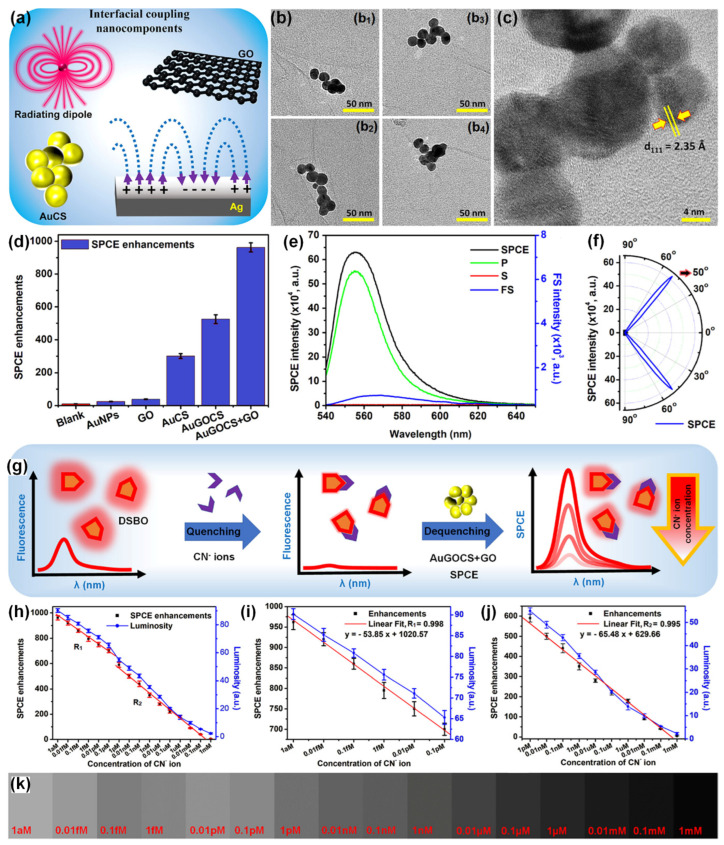
(**a**) Interfacial coupling nanocomponents including AuCSs, GO, radiating dipoles, and Ag metallic thin film. (**b**) Multiple TEM images (**b1**–**b4**) of AuGOCS indicating the GO flakes intertwined in and around the AuNPs in a single AuGOCS. (**c**) HRTEM image of AuGOCS displaying the lattice fringes of AuNPs. (**d**) SPCE enhancements obtained experimentally using different NPs and nanohybrid systems. (**e**) SPCE, FS, p and s intensity spectra for the sample presenting the highest SPCE enhancements in (**d**), i.e., for the AuGOCS + GO hybrid system. (**f**) Sharply directional emission observed experimentally with the AuGOCS + GO hybrid system, confirming the plasmon-coupled emission phenomena. (**g**) Conceptual schematic of the quenching with CN^−^ ions and dequenching with AuGOCS + GO in the SPCE platform. Cyanide ion sensing using AuGOCS + GO in the SPCE platform demonstrating the SPCE enhancements (left y-axis) and luminosity values (right y-axis) (**h**) over the complete sensing range from 1 aM to 1 mM, (**i**) over the sensing range R1 (from 1 aM to 0.1 pM), and (**j**) over the sensing range R2 (from 1 pM to 1 mM). (**k**) Shade cards corresponding to the luminosity values obtained using the smartphone-based detection platform. Adapted with permission from [49].

**Figure 12 nanomaterials-14-00111-f012:**
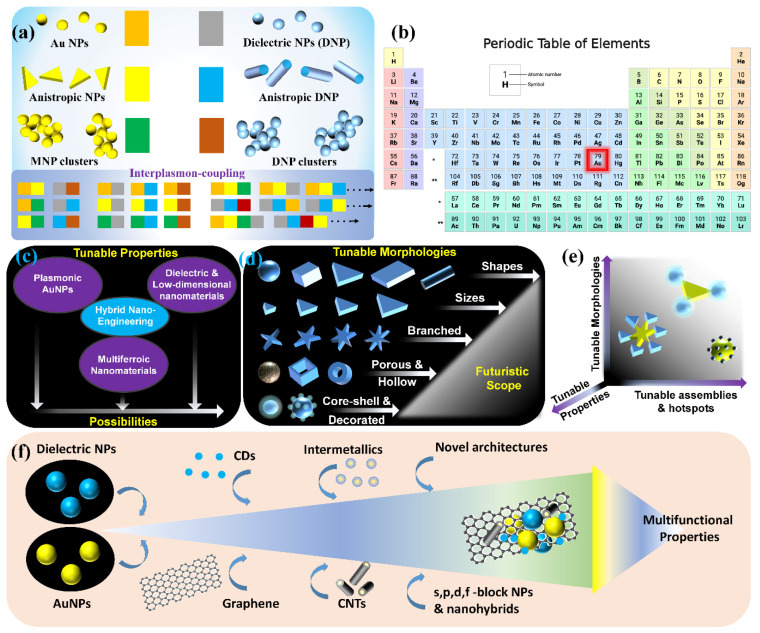
Futuristic scope and possibilities rendered by the nano-engineering technologies for the generation of different combination of plasmonic Au with isotropic and anisotropic dielectric nanohybrids (**a**,**c**–**e**). (**b**) The location of the plasmonic Au is highlighted in the periodic table. (**f**) A schematic representation of multifunctional properties that can be achieved using different combination of materials (such as metal–dielectric, low-dimensional substrates, sharp nano-architectures to name a few) at micro–nano regime. Adapted with permission from [42,55].

## Data Availability

Data are contained within the article.

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
