# Peer review of "Review of Gold Nanoparticles in Surface Plasmon-Coupled Emission Technology: Effect of Shape, Hollow Nanostructures, Nano-Assembly, Metal–Dielectric and Heterometallic Nanohybrids"

_nanomaterials, 2024, doi:10.3390/nano14010111_

Round 1
Reviewer 1 Report
Comments and Suggestions for Authors
1. The keywords “shape” and “heterometallic” are not suitable.
2. How do the authors think the relationship between “shape” and “hollow” in “differently engineered AuNPs in terms of shape, hollow, metal-dielectric, nano-assembly and heterometallic hybrids over the SPCE platform.”?
3. What does “AuSil” mean? When abbreviating any terms, spell them out the first time, and use the abbreviation after that.
Author Response
Please find the response attached herewith.

Reviewer 2 Report
Comments and Suggestions for Authors
This manuscript comprehensively reviews the nano-engineering and biosensing applications of AuNPs based on the shape, hollow morphology, metal-dielectric, nano-assembly and heterometallic nanohybrids. The manuscript is well-written and logically structured. Albeit that several points discussed by the authors are of great interest, some minor concerns have to be addressed before considering it for publication.
1. Considering the aspect of nanoengineering, establishing structure-property relationships is important for rational design. To properly achieve this, it is crucial to control the synthesis such that only a single particle parameter such as the shape is different while all other parameters such as size, capping agent etc are kept the same. This perquisite is still a challenge in the field of rational design of nanomaterials. Authors should discuss this point considering previous works e.g. 10.1002/adfm.202210945, 10.1021/acsami.7b07793.
2. Authors are advised to compare in a table the performance of some of the considered cases with respect to their sensitivity or enhancement factor.
3. May be the authors can think of shortening the title and summarizing the effects in one general aspect. But this is just a suggestion, and it is ok to keep the used one.
4. The introduction on nanoparticles-based sensors and fluorescence should be extended by including other works e.g. 10.1021/acs.jpclett.7b02925, 10.3390/biology11050621, 10.1039/D3RA05070J, 10.1021/acssensors.7b00382
5. The effect of capping agent on the properties of the nanomaterials can be discussed such as in ( Tanic acid capped gold nanoparticles: capping agent chemistry.. )
Author Response

(The authors gave the same response as above.)

Reviewer 3 Report
Comments and Suggestions for Authors
Comments on the manuscript
The manuscript reviews the use of gold nanoparticles (AuNPs) in Surface Plasmon Coupled Emission (SPCE) technology for point-of-care diagnostic platforms. It highlights the significance of SPCE in biosensing, emphasizing the advantages of AuNPs, such as stability, bio-functionalization feasibility, and photo-plasmonic response. The review covers various aspects, including nano-engineering, shape, hollow morphology, metal-dielectric, nano-assembly, and heterometallic nanohybrids of AuNPs in SPCE. The manuscript explores the principles of plasmonics, SPCE, and metal-enhanced fluorescence, providing case studies and discussing advancements. It concludes by emphasizing SPCE's potential for biosensing and its role in improving sensitivity in optical sensors. The comprehensive insights provided make it a valuable resource for researchers in photo-plasmonics and SPCE.
It is very impressive that the authors covered such a broad range of topics in a succinct manner. This review will be of interest to researchers in this field and beginners who are about to get into this research field. Consequently, I support the publication of this review after a minor revision. Below are my suggestions and comments.
1. What makes this review outstanding? Recent publications have extensively covered the subject of gold nanoparticle in LSPRs. How does this review stand out? Your insights are vital; kindly provide comments to articulate and emphasize our unique contribution in the introduction section.
2. I would like to draw the authors’ attention to the potential oversight regarding the discussion on the high quality factor (Q-factor) resonances in plasmonic systems. The significance of high-Q resonances in enhancing the performance of plasmonic structures, such as achieving giant field enhancement, is a noteworthy aspect that could contribute to the overall depth of your manuscript. I recommend considering the inclusion of recent works that emphasize the importance of high-Q plasmonics, particularly: "Bound states in the continuum in anisotropic plasmonic metasurfaces" (Nano Letters, 2020, 20(9), 6351-6356), which presents the initial experimental observation of plasmonic high-Q resonance in all gold split ring resonator arrays with significant field enhancement at the critical coupling condition. Also, "Laser‐Printed Plasmonic Metasurface Supporting Bound States in the Continuum Enhances and Shapes Infrared Spontaneous Emission of Coupled HgTe Quantum Dots" (Advanced Functional Materials, 2023, 33(44), 2307660), highlighting active plasmonic resonances for infrared spontaneous emission enhancement with coated quantum dots.
3. In Figure 1, could you elaborate on the key features illustrating the interaction between radiating dipoles and plasmonic NPs, metal thin film, and their combination, ensuring a clear understanding of the fundamental concepts presented in the manuscript?
4. When discussing the different architectures for embedding AuNPs, such as spin coating and electrostatic adsorption, could you elaborate on the specific advantages of each method, aiming to understand the distinct interactions between surface plasmons and fluorophores resulting from multiple architectures and their applications?
5. Can you provide insights into the specific linear relationship established between the AuNPs-enhanced SPCE signal and goat IgG concentration and its contribution to the overall performance of the immuno-sensor, particularly in terms of detection sensitivity, range, and limit?

readable
Author Response

(The authors gave the same response as above.)
